# Systematics of *Thraupis* (Aves, Passeriformes) reveals an extensive hybrid zone between *T. episcopus* (Blue-gray Tanager) and *T. sayaca* (Sayaca Tanager)

**Diego Cueva**[1,2¤]*, **Gustavo A. Bravo**[1,2,3], **Luís Fábio Silveira**[1]

**1** Museu de Zoologia da Universidade de São Paulo, São Paulo, SP, Brazil, **2** Sección de Ornitología, Colecciones Biológicas, Instituto de Investigación de Recursos Biológicos Alexander von Humboldt, Claustro de San Agustín, Villa de Leyva, Boyacá, Colombia, **3** Museum of Comparative Zoology and Department of Organismic and Evolutionary Biology, Harvard University, Cambridge, Massachusetts, United States of America

¤ Current address: Museum of Natural Science and Department of Biological Sciences, Louisiana State University, Baton Rouge, Louisiana, United States of America
* dacuevac@alumni.usp.br

**Data Availability Statement:** All relevant data are available on Figshare: https://doi.org/10.6084/m9.figshare.21084934.v1 https://doi.org/10.6084/m9.figshare.21084937.v1.

## Abstract

The Neotropical avian genus *Thraupis* (Passeriformes, Thraupidae) currently comprises seven species that are widespread and abundant throughout their ranges. However, no phylogenetic hypothesis with comprehensive intraspecific sampling is available for the group and, therefore, currently accepted species limits remain untested. We obtained sequence data for two mitochondrial (ND2, cyt-b) and three non-coding nuclear (TGFB2, MUSK, and βF5) markers from 118 vouchered museum specimens. We conducted population structure and coalescent-based species-tree analyses using a molecular clock calibration. We integrated these results with morphometric and coloration analyses of 1,003 museum specimens to assess species limits within *Thraupis*. Our results confirm that *Thraupis* is a monophyletic group and support its origin in the late Miocene and subsequent diversification during the Pleistocene. However, we found conflicts with previous phylogenies. We recovered *Thraupis glaucocolpa* to be sister to all other species in the genus, and *T. cyanoptera* to the remaining five species. Our phylogenetic trees and population structure analyses uncovered phylogeographic structure within *Thraupis episcopus* that is congruent with geographic patterns of phenotypic variation and distributions of some named taxa. The first genetic and phenotypic cluster in *T. episcopus* occurs east of the Andes and is diagnosed by the white patch on the lesser and median wing coverts, whereas the second group has a blue patch on the wing and distributes to the west of Colombia's eastern Andes. Finally, we present evidence of hybridization and ongoing gene flow between several taxa at different taxonomic levels and discuss its taxonomic implications.

**Funding:** We received a total of seventh sources of funding that we want to thank: DC 2015/22981-9 and 2017/03900-3 FAPESP – Fundação de Amparo à Pesquisa do Estado de São Paulo https://fapesp. br/en No The funders had no role in study design, data collection and analysis, decision to publish, or preparation of the manuscript. DC No grant number The Frank M. Chapman Memorial Fund of the American Museum of Natural History https:// www.amnh.org/research/vertebrate-zoology/ ornithology/grants No The funders had no role in study design, data collection and analysis, decision to publish, or preparation of the manuscript. GAB 2012/23852 FAPESP – Fundação de Amparo à Pesquisa do Estado de São Paulo https://fapesp.br/ en No The funders had no role in study design, data collection and analysis, decision to publish, or preparation of the manuscript. LFS 2017/23458-2 and 2018/20249-7 FAPESP – Fundação de Amparo à Pesquisa do Estado de São Paulo https://fapesp. br/en No The funders had no role in study design, data collection and analysis, decision to publish, or preparation of the manuscript. LFS 308337/2019-0 Brazilian Research Council https://www.gov.br/ cnpq/pt-br No The funders had no role in study design, data collection and analysis, decision to publish, or preparation of the manuscript.

**Competing interests:** The authors have declared that no competing interests exist.

# Introduction

The genus *Thraupis* (Aves, Thraupidae) is widely distributed across the Neotropics from northern Mexico to the Pampas of Argentina. All species are common and easily observed within their distributions, which include open areas, grasslands, scrub vegetation, crops, settlements, second-growth forests, and even urban areas of densely populated cities. Currently, *Thraupis* sensu stricto comprises seven species [1, 2]: *T. glaucocolpa* (Glaucous Tanager), *T. cyanoptera* (Azure-shouldered Tanager), *T. abbas* (Yellow-winged Tanager), *T. ornata* (Golden-chevroned Tanager), *T. palmarum* (Palm Tanager), *T. sayaca* (Sayaca Tanager), and *T. episcopus* (Blue-gray Tanager). Specimens of *Thraupis* are numerous in natural history museums, permitting a proper evaluation of inter- and intra-specific geographic variation of coloration. Moreover, across the lowlands of South and Central America at least one species of this genus occurs, making it an excellent model for investigating evolutionary and biogeographic questions [3], such as the effect of the Andes and other geographic barriers on speciation and color evolution [e.g., 4–8]. However, an objective and testable phylogenetic hypothesis and a robust definition of species limits are fundamental to address some of these questions.

The taxonomic history of *Thraupis* has been dynamic and convoluted with subspecies shifting between species, species between genera, and taxa changing from subspecies to species level and vice versa [9, 10]. Recent changes included the movement of two species into resurrected genera–*Rauenia bonariensis* (Blue-and-yellow Tanager) and *Sporothraupis cyanocephala* (Blue-capped Tanager)–because molecular phylogenetic evidence showed that they are not closely-related to the remainder species of *Thraupis* [2, 11–13]. Current phylogenetic evidence suggest that *Thraupis* is a monophyletic group that is closely related to the genera *Chalcothraupis*, *Poecilostreptus*, *Stilpnia*, *Tangara*, and *Ixothraupis*. Nonetheless, the latest available phylogenetic hypothesis of the genus lacks statistical support at some inner nodes, yielding unresolved positions for some species, such as *T. cyanoptera* [14]. Moreover, *T. glaucocolpa*, a species often considered as sister to *T. sayaca* due to coloration similarities, remains unsampled [10, 14].

Most taxonomic problems within *Thraupis* are concentrated within the group formed by *T. episcopus* and *T. sayaca*, hereafter referred to as the *episcopus-sayaca* complex. This complex contains a total of 17 subspecies (*T. episcopus episcopus*, *T. e. cana*, *T. e. caesitia*, *T. e. cumatilis*, *T. e. nesophila*, *T. e. ehrenreichi*, *T. e. berlepschi*, *T. e. quaesita*, *T. e. leucoptera*, *T. e. mediana*, *T. e. coelestis*, *T. e. caerulea*, *T. e. major*, *T. e. urubambae*; *T. sayaca sayaca*, *T. s. boliviana* and *T. s. obscura*) classified within the two species [15]. Moreover, due to its plumage similarity, *T. glaucocolpa* has historically been considered allied to the *episcopus-sayaca* complex. After its description, the placement of *T. glaucocolpa* has oscillated between species and subspecies, albeit always assumed to be closely related to *T. sayaca* [2, 10, 13, 14, 16]. Hilty [17] suggested that it is best treated as a superspecies with *T. episcopus* and *T. sayaca*, but its overlapping distribution with *T. episcopus* makes this relationship unlikely [2].

The Blue-gray Tanager (*T. episcopus*) is the species with the largest number of named subspecies, which are recognized based on slight variations of the wing-patch coloration, and blue tones on the chest and back. However, several subspecies have poorly defined geographic limits and are almost unidentifiable. Two major geographic groups are distinguishable within *T. episcopus* [see 18] based on the coloration of the lesser and median wing coverts (hereafter "wing-patch"), which ranges from white to ultramarine, with some populations showing sky blue and flax flower blue wing-patches [19]. All individuals with white wing-patch are found east of the Andes, whereas most taxa with blue wing-patch are west of the eastern Colombian Andes–including the Cauca and Magdalena valleys, and northern Colombia. However, there

are some individuals with intermediate phenotypes in the eastern Andes of Colombia with mixed blue and white wing-patches. Thus, population structure and taxonomic limits within the polytypic *T. episcopus* have yet to be thoroughly assessed [17, 18]. Several hybridization events have been reported between congeners: namely *T. episcopus* x *T. palmarum*, *T. episcopus* x *T. ornata*, and *T. episcopus* x *T. sayaca* [20]. In areas where *T. sayaca* and *T. episcopus* overlap, such as northwestern Bolivia and southeastern Peru [21], the correct identification of either species is extremely challenging because they tend to be phenotypically like one another, more so than in other parts of their ranges. Whether these instances of introgressive hybridization are widespread in those areas of parapatry or it is occasional, remains to be assessed.

Hybridization and introgression are important factors that affect evolution, diversification, and speciation processes. They can lead to different outcomes such as boostering speciation by reinforcement [22], merging taxa, generating new reticulate lineages [23, 24] or even transferring advantageous alleles from one lineage to another [25]. However, the underlying mechanisms leading to either outcome remain partially unclear. Hybridization often occurs in nature between closely-related species, and it has been widely reported in various bird families within songbirds (Passeriformes) [26–32]. Currently, one of the reasons that can lead to hybridization is human-induced habitat transformation [33, 34], which in the *episcopus-sayaca* complex might be relevant. Both species are largely parapatric and meet along the ecotone between Amazonia and the drier habitats in the Cerrado and Caatinga. Because Amazonian limits are receding due to deforestation, the limits and interactions between species may have been changing.

Here, we integrate molecular and morphological data to assess the current molecular hypothesis of the genus *Thraupis*. The integration of genotypic, and phenotypic information have led to major changes in systematic and taxonomic classifications, furthering the knowledge about the taxonomic limits across many bird groups [35–41]. Even in the genomic era, the integration of phylogenies based on few loci with morphological and bioacoustics analyses have identified evolutionary lineages with few external morphological differences and proposed novel biogeographical hypotheses [42–47]. Here, we infer a species-level phylogeny of the genus *Thraupis*, evaluate taxonomic boundaries within the *episcopus-sayaca* complex, and discuss the implications of interspecific introgressive hybridization for the taxonomy and evolution of *Thraupis*.

## Materials and methods

### Ethics statement

All newly collected specimens were obtained under regulations of the Brazilian National Council for the Control of Animal Experimentation (Conselho Nacional de Controle de Experimentação Animal–CONCEA; protocol (#001/2016) and Sistema de Autorização e Informação em Biodiversidade, SISBIO (#10013–4).

### Genetic sampling

We sampled a total of 113 vouchered specimens of *Thraupis* housed at eleven scientific collections (S1A Table in S1 File). We included all species in the genus, with emphasis on the *episcopus-sayaca* complex (15 of 17 subspecies: Fig 1). The distribution of *T. e. ehrenreichi* is limited to Huitanaã, on the Purús River [15, 48]. However, specimens analyzed from Acre, Brazil, match the original description of the subspecies and we consider these samples to represent *ehrenreichi*. Additionally, because the type locality of *T. e. major* is from Junín, Peru and we had no samples from that locality, we consider that subspecies unsampled. Likewise, we did not have access to samples of *T. sayaca boliviana*. Finally, five samples were included as

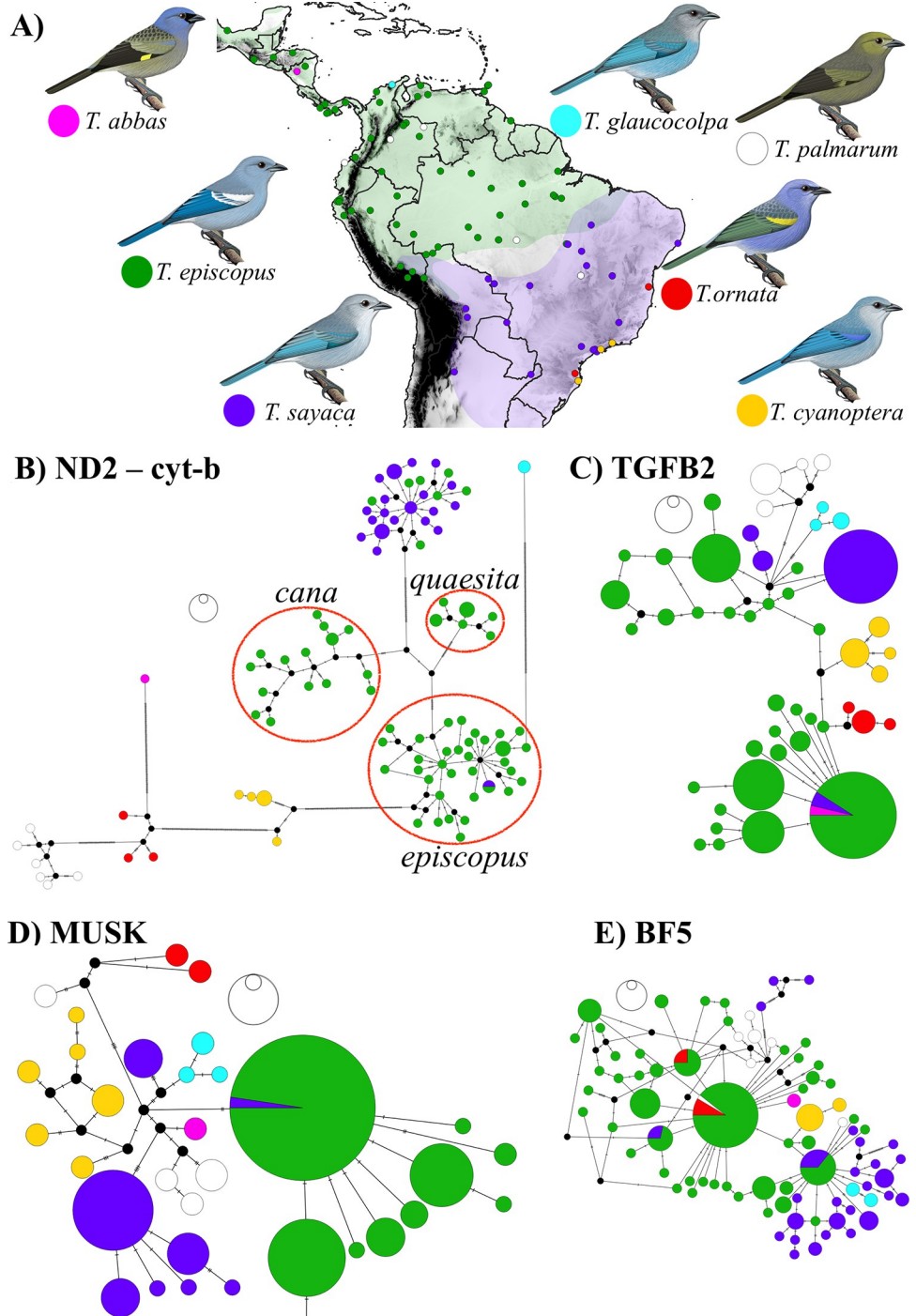

**Fig 1. Location and haplotype networks of 112 genetic samples of *Thraupis*.** (A) Taxonomic identification and geographic distribution of the genetic samples of *Thraupis* included in this study. Green and lilac areas show the distribution of *T. episcopus* and *T. sayaca*, respectively. Haplotype network of (B) the mitochondrial markers ND2 and cyt-b, (C) TGFB2, (D) MUSK, and (E) BF5 from the tissues referenced in Fig 1A. Red circles in the mitochondrial haplotype network highlight haplotype clusters and were named after the taxa with nomenclatural priority in the cluster but do not represent a single subspecies (i.e., the *episcopus* group includes several subspecies). Maps in this figure were made using the free software Qgis v.3.10.7 and free data layers from DIVA-GIS (https://www.diva-gis.org/) and distribution polygons from IUCN red list webpage [53, 54]. Tanager illustrations made by Fernando Ayerbe-Quiñones.

outgroups (*Paroaria baeri*, *Tangara chilensis*, *Tangara mexicana*, *Ixothraupis punctata*, and *Stilpnia cayana*) for a total of 118 samples. Outgroups were selected based on previously published phylogenies [7, 13].

We extracted total genomic DNA using the Genomic DNA Mini Kit (Invitrogen) and performed PCR amplification of the autosomal nuclear introns beta-fibrinogen intron 5 (BF5) and transforming Growth Factor Beta 2 intron 5 (TGFB2), the Z-linked muscle receptor tyrosine kinase (MUSK), and the mitochondrial markers cytochrome b (cyt-b) and NADH dehydrogenase 2 (ND2). Primers and protocols are summarized in S1B Table in S1 File. We assembled sequences using references from GenBank [49]. We used Geneious v.9.1.8 [50] to clean low-quality fragments, look for ambiguities, and align sequences using the MUSCLE algorithm with a maximum number of seven iterations. We phased nuclear sequences with heterozygous sites using Seqphase [51, 52]. We deposited sequences in GenBank with accession numbers ON552715 –ON552831, ON552832 –ON552949, OP289858 –OP289974, OP289975 –OP290081 and OP290082–290198. Of the 118 individuals, 15 did not produce sequence data for all five markers (i.e., BF5, TGFB2, MUSK, cyt-b, ND2). The total concatenated alignment length was 3,723 base pairs (bp) and most of the sequences yielded high quality reads for over 95% of the sequence length.

### Gene tree and population structure analyses

We built gene trees for each locus using RAxML v.8 [55] in the Cipres Science Gateway site v.3.2 [56]. For the mitochondrial locus (concatenated ND2 and cyt-b) we used Partition Finder2 v.2.1.1 [57] to select the best partition scheme. For all loci, we applied the nucleotide substitution model GTR + Γ and a rapid-bootstrap analysis using 999 replicates to assess nodal support. Also, for each locus, we constructed a median-joining haplotype network using the software PopART [58]. Informed by gene-tree and haplotype network analyses, we used the software STRUCTURE v. 2.3.3 [59] on the concatenated sequences (5 markers: 3,723 bp) of 95 samples (including phased nuclear sequences) within the *episcopus-sayaca* complex. We ran analyses by varying the number of putative populations (*k*) from 2 to 6 and running 10 iterations per each *k* value. We ran each Markov Chain Monte Carlo (MCMC) for a total of 50,000 generations, with a burn-in of 2,500. For each value of *k*, we selected the outcome with the highest likelihood and applied the method of Evanno et al. to assess the number of ancestral populations (best supported K) [60].

### Species tree analysis

To infer a species phylogeny, we conducted a Bayesian coalescent species tree analysis calibrated with a molecular clock using *BEAST [61, 62] on Cipres [56]. Taxa were defined *a priori* guided by results of network, gene-tree, and population structure analyses. Therefore, we used traditionally recognized species-level taxa as currently defined [2, 13], except for *T. episcopus*, which was separated into three groups. Also, to avoid violations of the assumption of no migration between species in the multispecies coalescent model [63, 64], we removed those individuals with hybrid ancestries in population structure analyses and intermediate genetic profiles (Table 1). Input data were prepared using BEAUti [65] within BEAST 2 v.2.4.6 [66]. Based on results from PartitionFinder2 [57], we chose GTR + Γ + I as the best substitution model for all loci except for ND2, which used the GTR + Γ substitution model. We ran three independent analyses with unlinked substitution models within partitions. For the mitochondrial locus, we used a strict clock with a clock rate of 0.0105 [67]. For the other three loci we used a log-normal relaxed clock, with the clock rate estimated from the data. We used the Yule model for tree shape and 200 million generations, sampling every 10,000 generations, and a

**Table 1. Specimens with gene tree conflicts between mitochondrial and nuclear markers.** Names in bold indicate specimens that clustered within a clade that differed from plumage-based assignments from museum identifications.

| CATALOGUE NUMBER | PLUMAGE-BASED ID | MUSK-Z CLADE | TGFB2 CLADE | ND2 –cyt-b CLADE |
|---|---|---|---|---|
| MZUSP 90269 | *T. sayaca* | *T. sayaca* | **T. episcopus** | *T. sayaca* |
| MZUSP 98621 | *T. sayaca* | **T. episcopus** | *T. sayaca* | **T. episcopus** |
| MZUSP (UFG4362) | *T. sayaca* | *T. sayaca* | **T. episcopus** | *T. sayaca* |
| KU 115635 | *T. episcopus* | *T. episcopus* | *T. episcopus* | **T. sayaca** |
| MZUSP 101479 | *T. episcopus* | *T. episcopus* | *T. episcopus* | **T. sayaca** |
| LSUMZ 9554 | *T. episcopus* | *T. episcopus* | *T. episcopus* | **T. sayaca** |
| MSB 27433 | *T. episcopus* | *T. episcopus* | *T. episcopus* | **T. sayaca** |
| MSB 36846 | *T. episcopus* | *T. episcopus* | *T. episcopus* | **T. sayaca** |
| MPEG T11969 | *T. episcopus* | *T. episcopus* | *T. episcopus* | **T. sayaca** |
| MZUSP 107246 | *T. episcopus* | *T. episcopus* | *T. episcopus* | **T. sayaca** |

burn-in of 25%. We combined log files with LogCombiner from BEAST2 v.2.4.6 [65] and used Tracer v.1.6.0 [68] to check that all ESS values were higher than 200. We also used LogCombiner to fuse the three species tree files and combine all the posterior probability into a single maximum clade credibility tree. We ran TreeAnnotator v.1.10 implemented in BEAST2 with a burn-in of 25% and a posterior probability limit of 0.5. We visualized tree figures with FigTree v.1.4.3 [69].

## Morphometrics

We measured a total of 1,003 specimens housed at six museum collections (S1C Table in S1 File) covering the geographic distribution of the complex [15]. Using a digital caliper, we measured total culmen (TC), culmen from nares to tip (CN), tarsus length (TS), wing chord (Wing), and tail length (Tail) following standard methods [70, 71]. When available, we obtained body weight from specimen labels. When coordinates were not specified, we used gazetteers to georeference specimens [72–75]. Because weight was the variable with the most missing data, we used a linear regression model between weight and the other measurements from specimens with full information to fill missing weight data. We excluded *T. glaucocolpa* from morphometric comparisons because it is not closely related to the *episcopus-sayaca* complex (see results). Moreover, comparisons were made grouping specimens into the resulting categories of the molecular analyses: one for *T. sayaca* and two within *T. episcopus*. To assess the variation in multivariate morphometric space, we performed a Principal Component Analysis (PCA), and to assess whether mean morphometric values among groups were significantly different, we conducted a Multivariate Analysis of Variance (MANOVA), with and without missing data replacement. Finally, we conducted a Linear Discriminant Analysis (LDA) to evaluate whether groups were morphometrically diagnosable.

## Coloration

We photographed 353 specimens that represent a subset of those used in morphometric analyses (S1D Table in S1 File), covering most of the geographic distribution and the color variation observed in the *episcopus-sayaca* complex. Also, we had access to photographs of type specimens, most of which were not taken within the following standardized conditions and were only used for visual comparisons (folder C available at figshare.com - S1 folder).

We used a Nikon D800 camera with a lens AF-S NIKKOR 28-300mm f/3.5–5.6G ED VR, a black background, and a grey standard. We standardized the angle of view by using a tripod

and placing specimens 35 cm from the lens. We used a shutter speed of 1/250, F 10.0, ISO 200, built-in flash -2.0, and white balance for flash +3.0 B6. Using Photoshop V. 19.0 and a gray card in each picture, we standardized white balance across pictures, making pictures comparable even with different light conditions across different collections. We extracted the HTML color code from the crown, chest, mantle, distal edging of the primary feathers, and the wing patch. We used the HTML code to color the georeferenced points of each specimen. Thus, we ended with a color distribution map for each of the four body parts. Lastly, we quantified the amount of white on the greater coverts with a categorical scale from 0 (no white) to 5 (the maximum amount of white found on *T. episcopus*) (S1 Fig).

## Results

### Phylogenetic relationships within *Thraupis*

Nuclear gene trees were poorly resolved but showed novel patterns (S2 Fig). For instance, in the TGFB2 and MUSK gene trees, most individuals clustered with conspecifics based on the morphological identification of voucher museum specimens. However, three individuals morphologically identified as *T. sayaca* clustered with individuals identified as *T. episcopus* (one bird in MUSK and two in TGFB2; S3 and S4 Figs). Unlike nuclear gene trees, the mitochondrial gene tree (ND2 and cyt-b) contained high bootstrap support for most relationships and clustered most samples with conspecifics (S5 Fig). Nevertheless, a total of eight individuals, seven *T. episcopus* and one *T. sayaca* were recovered within the other species (Table 1). Additionally, the mitochondrial gene tree recovered T. *glaucocolpa* as the sister species of all other members in the genus, and the remaining *Thraupis* species form three clades with unresolved relationships among them. The first clade includes *Thraupis cyanoptera* only. The second clade comprises *Thraupis abbas*, *T. ornata* and *T. palmarum*, whereas the third clade is formed by *T. sayaca* and *T. episcopus* (S5 Fig). Moreover, mitochondrial markers revealed a geographic pattern of genetic structure within *T. episcopus*, with samples clustering into three groups (Fig 1B and S5 Fig), hereafter referred to as the "*episcopus*," "*cana*" and "*quaesita*" groups. The *quaesita* group was recovered as the sister of the other two. The group with the most subspecies and wider distribution is *episcopus*, which included samples from Amazonia, Llanos, and Trinidad and Tobago. Its distribution coincides with the nominate subspecies *episcopus* and the subspecies *coelestis*, *leucoptera*, *major*, *nesophila*, *mediana*, *urubambae*, *berlepschi*, and *caerulea*. The *cana* group includes populations from the Magdalena and Cauca valleys and the adjacent Andean slopes, northern Colombia, Central America, and nearby islands, and coincides with the subspecies *cana*, *cumatilis*, and *caesitia*. The last group, *quaesita*, contains only individuals of the subspecies *quaesita*, which is found in the Pacific lowlands of western Colombia, Ecuador, and northwestern Peru.

Population structure and species tree analyses recovered the same three genetic groups within *T. episcopus* but with *cana* and *quaesita* as sister taxa (Fig 2A and 2B) recognizing as many as four populations in the *episcopus-sayaca* complex. However, the Evanno et al.'s method suggests that the maximum number of individual populations is three (best supported K = 3) [60], merging *cana* and *quaesita* groups into a single cluster, which are the most similar groups based on plumage. Similarly, the species tree analysis also recovered *cana* and *quaesita* as sister lineages.

The species tree analysis recovered *T. glaucocolpa* as sister to all other congeners, as the mitochondrial gene tree, and *T. cyanoptera* as sister to the remaining species in the genus. Also, the species tree suggested that *Thraupis* emerged between 5 and 5.5 million years ago at the Miocene–Pliocene boundary. Finally, all current lineages diversified between the Pliocene and the Pleistocene.

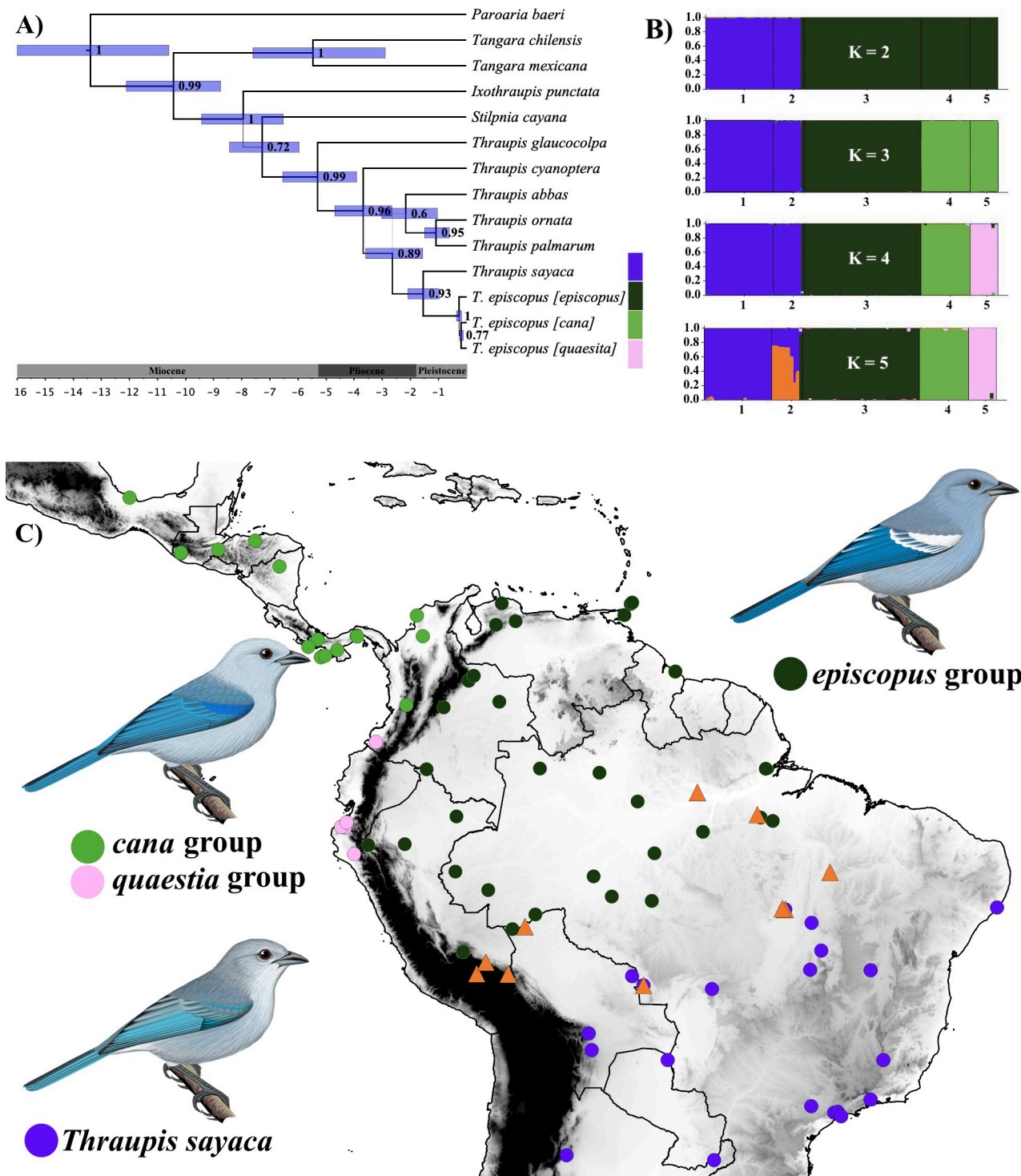

**Fig 2. Phylogenetic relationships within *Thraupis* and geographic distribution of the genetic structure within the *episcopus-sayaca* complex.** (A) Phylogenetic relationships of *Thraupis*. Names in brackets represent subspecies groups, with the name corresponding to the taxon with nomenclatural priority within each genetic cluster (i.e., a single group can include several subspecies). Thickness of the branches is associated with the posterior probability value on the corresponding node. (B) STRUCTURE results of the *episcopus-sayaca* complex, specimens in group 2 represents the putative hybrids (Table 1), best supported K = 3. (C) Geographic distribution of the *episcopus-sayaca* complex. Colored circles represent different lineages in the species tree and groups in the STRUCTURE plots. Orange triangles represent genotypically inconsistent specimens (Table 1), presumed to be hybrids between *episcopus* and *sayaca*. Color continuum from withe to black represent changes in elevation (withe = 0 m, black ≥ 2500 m). Maps in this figure were made using the free software Qgis v.3.10.7, free data layers from DIVA-GIS (https://www.diva-gis.org/). Tanager illustrations by Fernando Ayerbe-Quiñones.

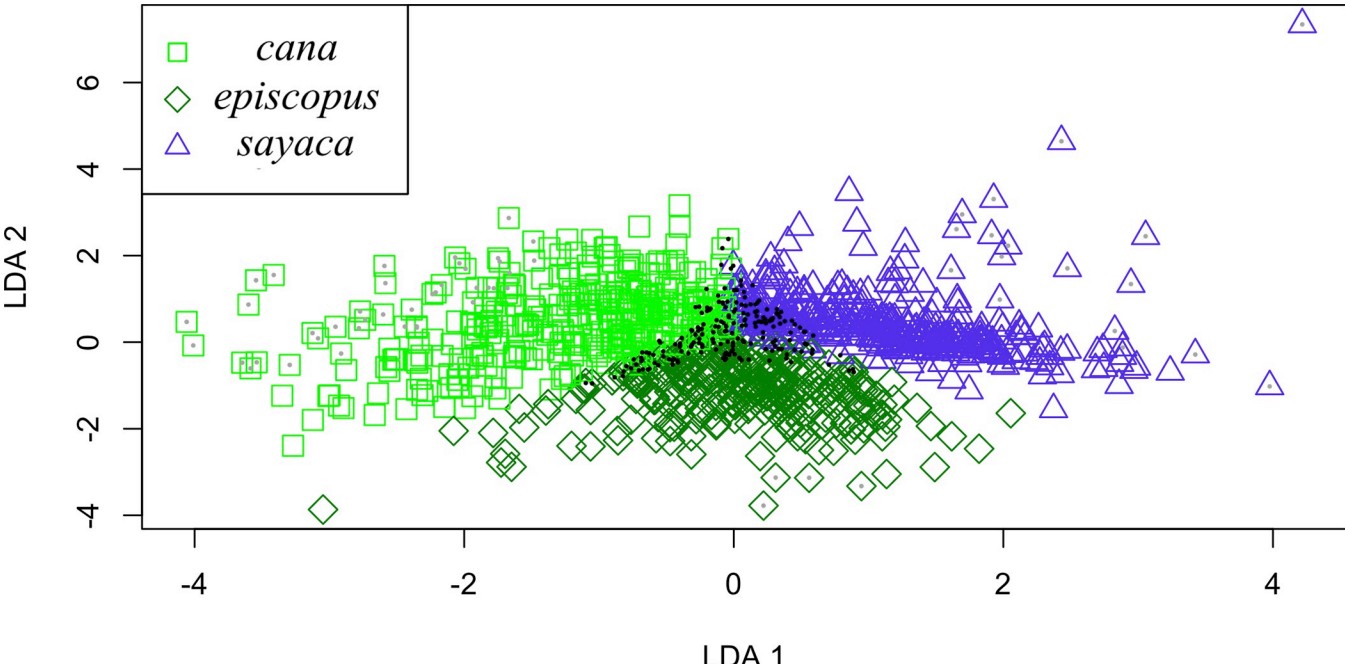

**Fig 3. Linear Discriminant Analysis (LDA) of morphometric traits within the *episcopus-sayaca* species complex.** Coefficients of linear discriminants for LDA1 are: -6.557557*Weight + 7.451838*CN + 1.987242*CT + 3.975692*TS + 16.869735*Wing + 2.702621*Tail. Coefficients of linear discriminants for LDA2 are: 14.460641*Weight + 1.987242*CN—4.708751*CT + 15.982253*TS—11.342798*Wing—1.458385*Tail. Individuals marked with gray dots were identified with a posterior probability of 0.9 or higher and represent 6.1% of the birds. Individuals marked with black dots were identified with a posterior probability of 0.5 or lower and represent 19.2% of the birds.

## Morphometrics

Following results of the population structure and the species tree analyses, we considered *T. sayaca* as a single group and divided *T. episcopus* into two groups for morphometric analyses: *episcopus*–east of the Andes–and *cana*–west of the eastern Andes. The principal component analysis did not show isolation between any of the groups (S6 Fig). However, we found that mean morphometric values are significantly different, with or without replacing missing data (P<0.001 in both cases). Similarly, linear discriminant analyses showed that all groups clustered together with only 6.1% of individuals identifiable with a 0.9 posterior probability or higher and 19.2% of the individuals identified with a posterior probability of 0.5 or lower. Most specimens clustered toward center of morphometric space (Fig 3).

## Coloration

The plumage analysis of the *episcopus-sayaca* species complex recovered three geographic patterns. Within *T. episcopus*, wing-patch color varies geographically. Birds west of the Andes have a blue wing-patch, whereas populations east of the Andes have a white wing-patch, except for populations from the Llanos in Colombia and Venezuela, and Trinidad and Tobago, which have white, blue, and intermediate wing-patches (Figs 4 and 5). Birds from Trinidad and Tobago have intermediate wing-patches but were more morphologically homogeneous than individuals from continental regions. Moreover, putative genetic intermediates (Table 1) between *T. episcopus* and *T. sayaca* also have intermediate wing-patch coloration. Within the *episcopus* group, the turquoise coloration of the chest and the crown is more iridescent and brighter in western Amazonia and becomes duller away from this region (S7 and S8 Figs),

followed by the decrease in the amount of white on the greater wing coverts (S9 Fig). We did not detect any clear geographic pattern in the coloration of the back and primary feathers (S10 and S11 Figs).

## Discussion

### A phylogenetic hypothesis for the genus *Thraupis*: A story of pervasive hybridization?

Our phylogenetic hypothesis uncovers key aspects of the relationships within *Thraupis* and confirms some previous phylogenetic assessments [7, 13, 14]. We corroborate that, as currently defined, *Thraupis* is monophyletic and we provide new insights into the phylogenetic position of previously unsampled taxa. We found that *T. glaucocolpa* is not sister or closely-related to *T. sayaca*, as previously thought [14], but rather it is the sister species to all species in the genus. This finding suggests that the plumage similarity between *T. glaucocolpa* and *T. sayaca*, which was interpreted as evidence of their close relationship, may reflect either an ancestral plumage pattern retained in both lineages or parallel evolution of plumage coloration.

Although the phylogenetic position of *T. glaucocolpa* and *T. cyanoptera* was well supported, we recovered *T. abbas* with a low posterior probability in the species tree. Possible explanations for this low resolution include incomplete lineage sorting or recent introgression [64] with other *Thraupis* species. Evidence for hibridization comes from the TGFB2 haplotype network, which nests *T. abbas* within *T. episcopus* at this locus (Fig 1C). Moreover, this possibility is further supported by a putative hybrid specimen *T. episcopus* x *T. abbas* (Fig 5I), housed at the Museum of Comparative Zoology (MCZ 163160), and collected at Comayagüela, Honduras, where the only two co-ccurring species are *T. episcopus* and *T. abbas*. This specimen is entirely purple with the head and back coloration of *T. abbas* (Fig 5J) but shows only a slight contrast between its head and body and does not have black marks on the lores, as in *T. episcopus*. Introgression can lead to low resolution across the phylogeny complicating the phylogenetic placement of some taxa. Genomic analyses will provide further insights into the phylogeny of the genus and will inform the extent of past on on-going hybridization between congeners [e.g., 76].

The critical role of hybridization in the evolutionary history of *Thraupis* is also evident between parapatric populations belonging to sister species. We found that *T. episcopus* and *T. sayaca* diverged ca. .5 million years ago and currently hybridize over a broad region of central South America, along the ecotone between Amazonia and the South American Dry Diagonal, where they are parapatric (S12 Fig). There are several specimens showing intermediate traits between the two species (Fig 5F and 5G) that also were confirmed as hybrids by gene trees and population structure analyses (Table 1). Of the ten hybrids reported here, two were collected in central Amazonia in Pará, Brazil (S12 Fig), far from the range of *T. sayaca*. It is possible that anthropogenic disturbances, such as deforestation, in the Amazon basin created similar environmental conditions to those found across the native range of *T. sayaca*, and generated opportunities for its dispersal into the distribution of *T. episcopus*. These hybrids were collected in 2009 in an area bisected by the Trans-Amazonian Highway (Rodovia Transamazônica), whose construction started in the early 1970's and is a powerful driver of deforestation in central Amazonia [77]. Vegetation transformation from primary forest to open and secondary edge vegetation may have favored the dispersal of *T. sayaca* into Amazonia. Further analyses based on genomic data will be critical to attain a comprehensive understanding of the history of hybridization along ecotones.

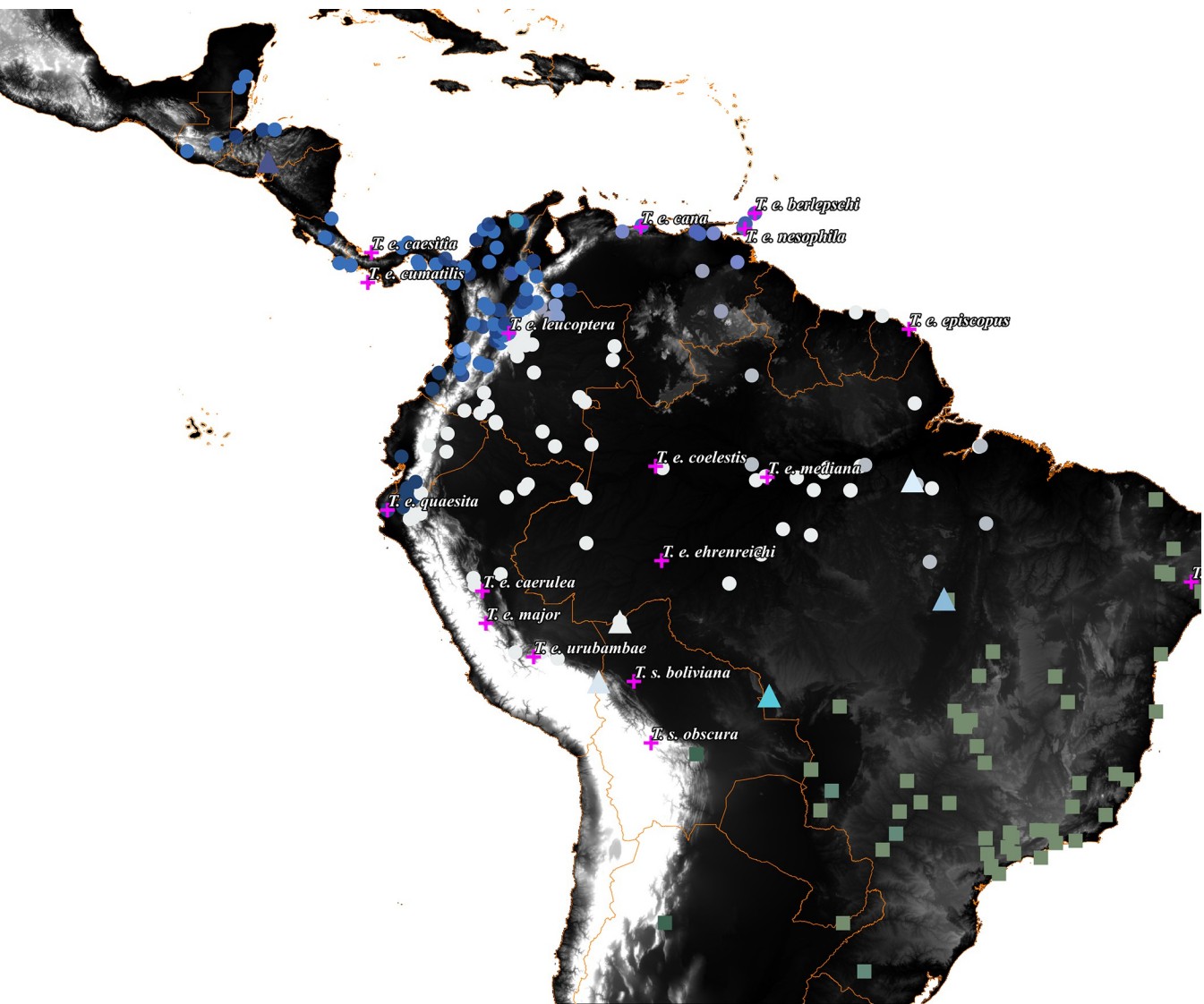

**Fig 4. Geographic distribution of wing-patch color in the *episcopus-sayaca* species complex.** Each symbol represents a photographed specimen of *T. episcopus* (circles), *T. sayaca* (squares), or intermediate specimens (triangles). Symbol colors denote the actual color of the wing patch, as extracted from photography with the HTML code. Pink crosses represent the type locality of each subspecies. Color scale from black to withe represent changes in elevation (black = 0 m, white ≥ 3000 m). Maps in this figure were made using the free software Qgis v.3.10.7 and free data layers from DIVA-GIS (https://www.diva-gis.org/).

## Comments on the taxonomy of *T. episcopus* and *T. sayaca*

Based on the unified species concept proposed by de Queiroz [78], which posits that species are "separately evolving metapolulation lineages" that can be delimited by several not obligatory or exclusive species criterion (i.e., reproductive isolation, phenotypically distinguishable, ecologically distinct to mention some), and the results presented herein, we suggest maintaining the species rank for *T. episcopus* and *T. sayaca*. Excluding hybrids, adults of both species are morphologically distinct and easily diagnosed by the body and wing-patch coloration (Fig 5). Moreover, these taxa were recovered as two well-supported groups in our phylogenetic and population structure analyses (Fig 2). They tend to occur in different ecosystems and hybrids seem to be restricted to the ecotone between Amazon and the South American Dry Diagonal.

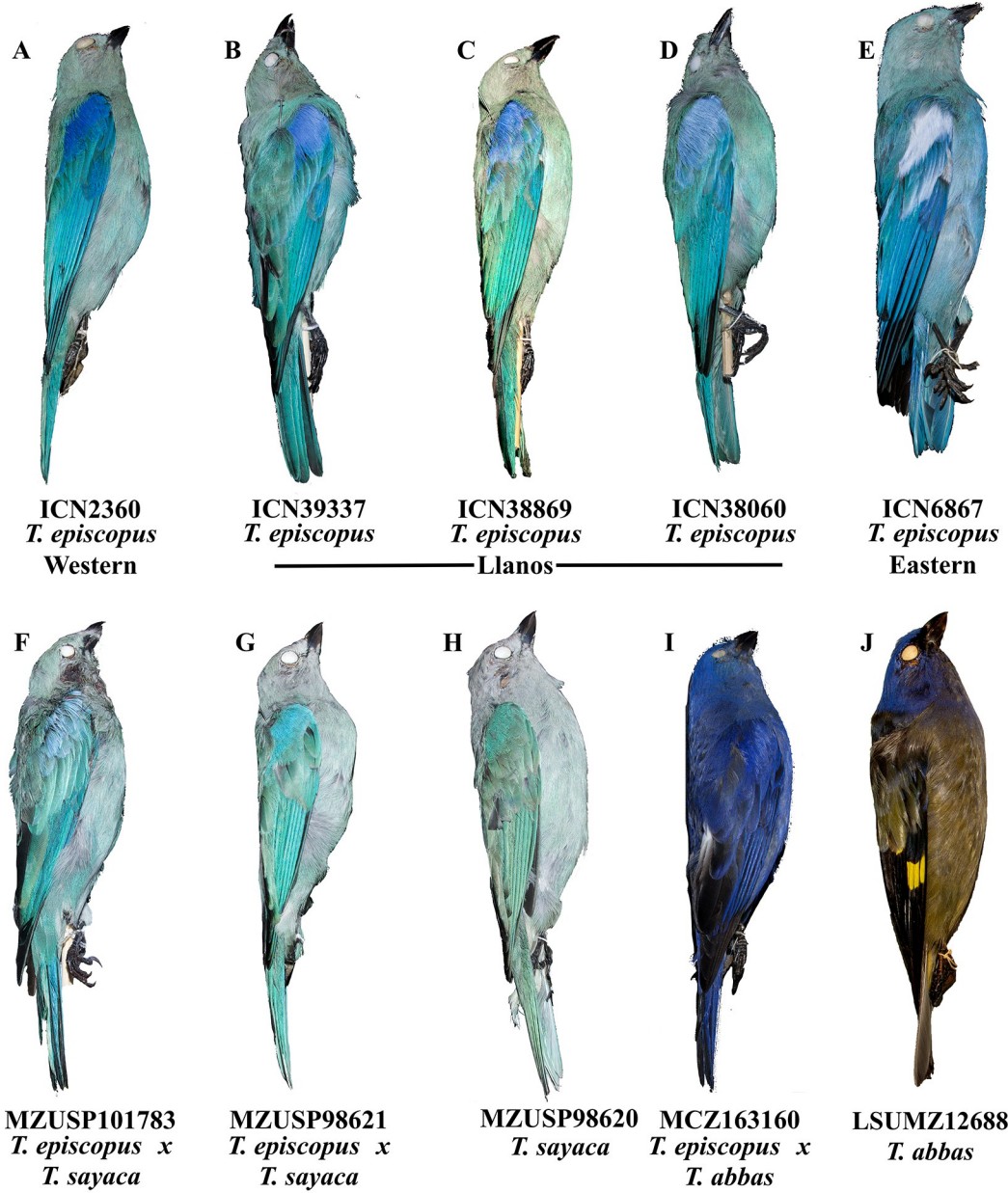

| | | | | |
|---|---|---|---|---|
| **A** | **B** | **C** | **D** | **E** |
| **ICN2360** | **ICN39337** | **ICN38869** | **ICN38060** | **ICN6867** |
| *T. episcopus* | *T. episcopus* | *T. episcopus* | *T. episcopus* | *T. episcopus* |
| **Western** | ——————— Llanos ——————— | | | **Eastern** |

| | | | | |
|---|---|---|---|---|
| **F** | **G** | **H** | **I** | **J** |
| **MZUSP101783** | **MZUSP98621** | **MZUSP98620** | **MCZ163160** | **LSUMZ12688** |
| *T. episcopus* x | *T. episcopus* x | *T. sayaca* | *T. episcopus* x | *T. abbas* |
| *T. sayaca* | *T. sayaca* | | *T. abbas* | |

**5 cm**

**Fig 5. Morphological variation in the *episcopus-sayaca* complex and a presumed *T. episcopus* x *T. abbas* hybrid.** (A) typical specimen of *T. episcopus* from the western side of the Andes. (B—D) intermediate individuals between western and eastern populations of *T. episcopus* in Colombia and Venezuela Llanos, showing the color variation of intermediate individuals. (E) Representative bird from eastern of the Andes, although this is the most morphologically variable group. (F—G) specimens of *T. episcopus* x *T. sayaca* morphological and molecular hybrids, with intermediate colors in the wing patch. (H). The typical phenotype of *T. sayaca*, and (I) presumed *T. episcopus* x *T. abbas* hybrid next to (J) a typical *T. abbas* individual.

At the subspecies level, we suggest taxonomic changes, especially within *T. episcopus*. First, we recognize two major groups within *T. episcopus*, one east of the Andes with almost all individuals exhibiting a white wing patch, and a second group west of the Andes with a blue wing

patch. We suggest these groups be maintained as a single species given the pervasive hybridization in the llanos of Colombia and Venezuela, as indicated by the common occurrence of intermediate birds showing the full spectrum of wing-patch colors, from white to blue, including violet and lilac, away from populations of the "parental" subspecies. Finally, both our phylogenetic and population structure analyses support the treatment of both wing-patch groups as a single species.

The group east of the Andes comprises a total of ten subspecies; eight of them–*T. e. episcopus* [9], *T. e. coelestis* [79], *T. e. leucoptera* [80], *T. e. major* [81], *T. e. ehrenreichi* [82], *T. e. caerulea* [83], *T. e. mediana* and *T. e. urubambae* [84]–have a white wing patch; and two–*T. e. nesophila* [85, 86] and *T. e. berlepschi* [87]–have a violet and ultramarine wing patch. We did not find a single discrete character that separates the white wing patch taxa from the nominal form. Instead, we found a clinal color variation (S7–S11 and S13 Figs, and folder A available at figshare.com - S1 folder) of the characters used in the original descriptions, namely the amount of white in the wing, the color of the wing patch, head, chest, and back. The rump color was a common character used in the description of the subspecies. However, it is commonly hidden under the wings of museum specimens, and it was difficult to observe in most specimens. Birds near the Napo region, in western Amazonia, are bluer and with some iridescent feathers on chest and head. They have a white wing patch, the maximum category of white in the greater wing coverts (S9 Fig), a turquoise head that contrasts with the back, a blue iridescent chest, and completely blue wings and tail. Specimens away from this region become duller and with less white on the wing patch and greater coverts (see supplemental material). Furthermore, the hypothesis that different subspecies are intermediate along a continuous cline comes from the original descriptions. Zimmer wrote about *T. episcopus mediana*: "This form is admittedly intermediate between *episcopus* and *coelestis*, but it is relatively consistent over a very extensive range and deserves recognition as a distinct form" and also reported: "There is more tendency toward integration between *mediana* and *episcopus* than between *mediana* and *coelestis* where the distinction is more abrupt" [84]. Here, we confirm that most subspecies in *T. episcopus* are points along a continuous cline that perhaps seemed discrete to past authors because of sampling gaps. Thus, because there are no discrete characters that permit the diagnosis of geographic populations from others, we suggest synonymizing all subspecies with white wing-patch into *T. e. episcopus*.

The taxa with violet and ultramarine colors on the wing patch, *T. e. nesophila* and *T. e. berlepschi*, are a special case. Within the continental specimens, it is possible to find birds with all intermediate colors of wing-patch between white and blue across the llanos of Colombia and Venezuela. These birds are classified as *T. episcopus nesophila*. However, because they are intermediate between two groups and its diagnostic character, the violet-blue wing-patch [85], is not stable, we suggest the continental group should not represent a separate taxon. On the other hand, most specimens from the islands of Trinidad and Tobago have a consistent morphology with almost completely ultramarine wing-patch and rump, and blue iridescent feathers on the chest, distinct from similar specimens from the mainland. This form should be considered a valid subspecies due to its discrete morphology and insular distribution. Since its original description in 1880 [85], the type locality of *T. episcopus nesophila* is assumed to be Trinidad [10], and so, it is the senior synonym of *T. e. berlepschi*. It is important to highlight that some specimens from Trinidad may exhibit violet colors on the wing-patch and specimens from the mainland may have similar colorations on the rump, chest, and wing-patch as the island group.

The second group is located west of the Andes and has a prevalent blue wing-patch, being formed by *T. episcopus cana* [88], *T. episcopus quaesita* [89], *T. episcopus cumatilis* [90] and *T. episcopus caesitia* [91]. Excluding *T. e. caesitia*–not included in our analyses–we did not find

any consistent character that distinguished these subspecies from *T. e. cana* (folder B available at figshare.com - S1 folder). However, because we found that phylogenetic and population structure analyses recovered *T. e. quaesita* as a distinct group (Fig 1B), we suggest maintaining this taxon until more comprehensive sampling across the Chocó rainforest is available. Thus, *T. e. cana* is the senior synonym of *T. e. cumatilis*, and we suggest maintaining *T. e. quaesita* and *T. e. caesitia*, given our incomplete sampling.

Finally, within *T. sayaca* [9], we did not analyze many specimens of the subspecies *T. s. obscura* [92] and *T. s. boliviana* [93]. However, the type specimen of *T. s. boliviana* was analyzed from high quality photos (Supplemental Material). This specimen was collected near the Bolivia–Peru border–a region where we confirmed four hybrids–and shows a wing patch with almost identical coloration of one of the hybrids we sequenced (Fig 5G). Thus, we consider *T. s. boliviana* to be intermediate between *T. episcopus* and *T. sayaca*, and, therefore, not a valid taxon.

Here, we showed that *Thraupis* is a genus comprised of seven species and originated at least 5 million years ago near the Miocene–Pliocene boundary. We found evidence that suggests that hybridization has been an important factor in *Thraupis* evolution, especially between closely related species such as *T. episcopus* and *T. sayaca*, and current anthropogenic activity as deforestation can be important factors that modify current hybridization dynamics. Based on our morphological and genetic data, we conclude that *T. episcopus* and *T. sayaca* should be treated as a separated species with a maximum of five subspecies. Finally, we want to emphasize the importance of scientific collections on this study and the importance of studying common and abundant species.

## Supporting information

**S1 Text. Alternative language abstract.** Abstract in Spanish.
(PDF)

**S1 Fig. Classification of the amount of white on greater wing coverts in *Thraupis episcopus*.** Illustration showing the difference of amount of white. Dash line denotes the main rachis of the greater coverts and the black area the amount of white on the feather. Values of 0 represent no white and 5 the maximum amount of white.
(PNG)

**S2 Fig. Maximum-likelihood gene tree of the nuclear marker BF5.** Raw tree files at figshare.com–S2 folder.
(PNG)

**S3 Fig. Maximum-likelihood gene tree of the nuclear marker MUSK.** Raw tree files at figshare.com–S2 folder.
(PNG)

**S4 Fig. Maximum-likelihood gene tree of the nuclear marker TGFB2.** figshare.com–S2 folder.
(PNG)

**S5 Fig. Maximum-likelihood gene tree of the concatenate mitochondrial markers ND2 and cyt-b.**
(PNG)

**S6 Fig. Principal component analyses of morphmetic traits within *Thraupis episcopus*.** (A) Loadings of the PCA. (B) PCA of the three main groups recovered in genetic analyses. The

names *episcopus*, *cana* and *sayaca* were given based on nomenclatural priority in each group. All comprise more than one named subspecies (see main S1 Text).
(PNG)

**S7 Fig. Geographic distribution of chest color variation in the *episcopus-sayaca* species complex.** Each symbol represents a photographed specimen of *T. episcopus* (circles), *T. sayaca* (squares) or intermediate specimens (triangles). Symbol colors represent the actual color of the chest, as extracted from photography with the HTML code. Pink crosses denote the type locality of each subspecies. Raw pictures available at figshare.com—S1 folder.
(PNG)

**S8 Fig. Geographic distribution of crown color variation in the *episcopus-sayaca* species complex.** Each symbol represents a photographed specimen of *T. episcopus* (circles), *T. sayaca* (squares) or intermediate specimens (triangles). Colors of the symbols reflects the actual color of the crown, as extracted from photography with the HTML code. Pink crosses mark the type locality of each subspecies. Raw pictures available at figshare.com—S1 folder.
(PNG)

**S9 Fig. Geographic distribution of amount of white on the wing greater coverts in the *Thraupis episcopus*.** Each white circle represents a photographed specimen of *T. episcopus*. The amount of white was estimated using S1 Fig. The size of the with circles represent the amount of white on the wing greater coverts. Individuals with category 0 or no white are not on the map. Raw pictures available at figshare.com—S1 folder.
(TIF)

**S10 Fig. Geographic distribution of back color in the *episcopus-sayaca* species complex.** Each symbol represents a photographed specimen of *T. episcopus* (circles), *T. sayaca* (squares) or intermediate specimens (triangles). Colors of the symbols reflects the actual color of the back, as extracted from photography with the HTML code. Pink crosses mark the type locality of each subspecies. Raw pictures available at figshare.com—S1 folder.
(PNG)

**S11 Fig. Geographic distribution of primaries color in the *episcopus-sayaca* species complex.** Each symbol represents a photographed specimen of *T. episcopus* (circles), *T. sayaca* (squares) or intermediate specimens (triangles). Colors of the symbols reflects the actual color of the primaries, as extracted from photography with the HTML code. Pink crosses mark the type locality of each subspecies. Raw pictures available at figshare.com—S1 folder.
(PNG)

**S12 Fig. Geographic distribution of molecular hybrids and deforestation path connecting to inner Amazon hybrids.** Pink triangles represent the locality of hybrid specimens in Table 1. Green and Purple represents the geographic distribution of *T. episcopus* and *T. sayaca* respectively [53, 54]. Black arrows indicate deforestation path that connect the Cerrado grasslands with the hybrid molecular specimens on the middle of the Amazon. Most of the highlighted deforestation path belongs to the Trans-Amazonian Highway.
(PNG)

**S13 Fig. Geographic distribution of *Thraupis episcous* subspecies and coloration examples of collected individuals (See pictures in S1 folder).** Pink crosses mark the type locality of each subspecies. Blue circles with numbers are link with the pictures on S1.
(PNG)

**S1 File.** A Sequenced specimen information; B Protocol and details of Polymerase Chain Reactions; C Morphometric data; D Photographed specimen information.
(XLSX)

## Acknowledgments

We thank all museums, institutions, curators and staff that permitted access to the ornithology collections of skins and tissues, either by loan or pictures in the case of type specimens: J. Weckstein, N, Rice (The Academy of Natural Sciences of Drexel University), J. L. Cracraft, T. J. Trombone, M. Shanley (American Museum of Natural History), R. Moyle, M. Robbins (Kansas University–Biodiversity Institute and Natural History Museum), M. Lentino (Colección Ornitológica Phelps), J. Bates, M. Hennen, S. Lyon (The Field Museum), S. V. Edwards, J. Trimble, K. Eldridge (Harvard University–Museum of Comparative Zoology), C. Gómez, L. Leyton, D. Ocampo (Instituto Alexander von Humboldt), J. V. Remsen Jr., S. Cardiff (Louisiana State University–Museum of Natural Sciences), H. Van Grouw (Natural History Museum), G. Mayr (Natural History Museum Frankfurt), A. Aleixo (Museu Paraense Emílio Goeldi), S. Frahnert (Museum für Naturkunde), R. Becker (Museum für Vogelkunde Heineanum), H. James, C. Gebhard, B. Schmidt (Smithsonian National Museum of Natural History), F. G. Stiles (Universidad Nacional de Colombia–Instituto de Ciencias Naturales), M. Lowe (The University Museum of Zoology, Cambridge), C. C. Witt (University of New Mexico–Museum of Southeestern Biology), H. Mejlon (Uppsala University–Museum of Evolution) S. Rohwer, R. Faucett, C. Wood, (Washington University–Burke Museum), and M. Unsöld (Zoologische Staatssammlung München). Also, we want to thank W. Lemos de Morais Neto (Fartura Agropecuária), M. Sharp (Agropecuária Morro Branco), and B. Ehlers (UPS Brazil), provided financial and logistical support, and granted access to some important localities for collecting specimens. Instituto Chico Mendes de Conservação da Biodiversidade (ICMBio) for issuing collecting grants in Brazil. We want to thank BIOMATTERS for offering a free license of the software Geneious to the countries affected by the Chikungunya and Zika pandemics. Also, we are thankful to Idea Wild for the donation of equipment for this research. We are particularly grateful to J. Battilana at MZUSP BioMol and S. Herke at the Louisiana State University Genomics Facility for assistance with molecular lab work. We appreciate the support from members of the Ornithology Groups at Universidad Nacional de Colombia–GOUN, OrlandoAcevedo, Jualian Avila, Diego Carantón, Laura Echeverri, Natalia Perez, Alejandro Pinto, Juan Ríos, Nabhí Romero, Museu de Zoologia da Universidade de São Paulo, Cristiane Apolinario, Renata Beco, Fernanda Bocalini, Sergio Bolivar, Bianca Matinata, Instituto Alexander von Humboldt, Sebastian Pérez, Socorro Sierra, Juliana Soto, and Robb Brumfield's Lab at Louisiana State University, Eamon Corbett, Glaucia del Rio, Anna Hiller, Andre Moncrieff, Marco Rego. We are thankful to Fernando Ayerbe-Quiñones for allowing us the use of his bird illustrations. Finally, we thank Dan Lane, J. V. Remsen Jr., Oscar Johnson, Rafael Sobral Marcondes, Alparslan Yildirim and two anonymous reviewers for their comments on this manuscript.

## Author Contributions

**Conceptualization:** Diego Cueva, Gustavo A. Bravo, Luís Fábio Silveira.

**Data curation:** Diego Cueva.

**Formal analysis:** Diego Cueva.

**Funding acquisition:** Diego Cueva, Gustavo A. Bravo, Luís Fábio Silveira.

**Investigation:** Diego Cueva, Gustavo A. Bravo.

**Methodology:** Diego Cueva, Gustavo A. Bravo.

**Project administration:** Diego Cueva.

**Resources:** Diego Cueva, Luís Fábio Silveira.

**Software:** Diego Cueva.

**Supervision:** Diego Cueva, Gustavo A. Bravo, Luís Fábio Silveira.

**Validation:** Diego Cueva, Gustavo A. Bravo, Luís Fábio Silveira.

**Visualization:** Diego Cueva, Gustavo A. Bravo, Luís Fábio Silveira.

**Writing – original draft:** Diego Cueva, Gustavo A. Bravo, Luís Fábio Silveira.

**Writing – review & editing:** Diego Cueva, Gustavo A. Bravo, Luís Fábio Silveira.

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
