## [Decision Letter · Decision Letter 0]

16 Feb 2022

PONE-D-21-37811Systematics of Thraupis reveals hybridization of Thraupis episcopus (Blue-gray Tanager) at multiple scalesPLOS ONE

Dear Dr. Cueva,

Thank you for submitting your manuscript to PLOS ONE. After careful consideration, we feel that it has merit but does not fully meet PLOS ONE’s publication criteria as it currently stands. Therefore, we invite you to submit a revised version of the manuscript that addresses the points raised during the review process.

We look forward to receiving your revised manuscript.

Kind regards,

Tzen-Yuh Chiang

Academic Editor

PLOS ONE

Journal Requirements:

“DC

2015/22981-9 and 2017/03900-3

FAPESP – Fundação de Amparo à Pesquisa do Estado de São Paulo

https://fapesp.br/en

No

DC

No grant number

The Frank M. Chapman Memorial Fund of the American Museum of Natural History

https://www.amnh.org/research/vertebrate-zoology/ornithology/grants

No”

3. We note that you have stated that you will provide repository information for your data at acceptance. Should your manuscript be accepted for publication, we will hold it until you provide the relevant accession numbers or DOIs necessary to access your data. If you wish to make changes to your Data Availability statement, please describe these changes in your cover letter and we will update your Data Availability statement to reflect the information you provide

4. We note that Figure 1, 2 and 4 in your submission contain map images which may be copyrighted. All PLOS content is published under the Creative Commons Attribution License (CC BY 4.0), which means that the manuscript, images, and Supporting Information files will be freely available online, and any third party is permitted to access, download, copy, distribute, and use these materials in any way, even commercially, with proper attribution. For these reasons, we cannot publish previously copyrighted maps or satellite images created using proprietary data, such as Google software (Google Maps, Street View, and Earth). For more information, see our copyright guidelines: http://journals.plos.org/plosone/s/licenses-and-copyright.

 a. You may seek permission from the original copyright holder of Figure 1, 2 and 4 to publish the content specifically under the CC BY 4.0 license. 

Reviewers' comments:

Reviewer's Responses to Questions

**Comments to the Author**

1. Is the manuscript technically sound, and do the data support the conclusions?

Reviewer #1: Yes

Reviewer #2: Yes

Reviewer #3: Partly

2. Has the statistical analysis been performed appropriately and rigorously? 

Reviewer #1: Yes

Reviewer #2: N/A

Reviewer #3: Yes

3. Have the authors made all data underlying the findings in their manuscript fully available?

Reviewer #1: Yes

Reviewer #2: Yes

Reviewer #3: No

4. Is the manuscript presented in an intelligible fashion and written in standard English?

Reviewer #1: Yes

Reviewer #2: Yes

Reviewer #3: Yes

5. Review Comments to the Author

Reviewer #1: The MS presents unique and novel data on the phylogenomics and the evaluation of the species in Thraupis genus. It also provides new insights in the distribution of the species and their boundaries with morphometric evaluations. Study was well design and the results were presented in detail. So I conclude that this MS could highly contribute to the current knowledge in the era of Thraupis.

Reviewer #2: In this study the authors investigated the systematics of Thraupis and the hybridization of Thraupis episcopus. The manuscript is very interesting and well-written. The study falls into the scope of Plos one, thus it could be published. Before this can be done, a minor revision should be made as specified below.

- I suggest changing the title. The authors found evidence of hybridization between T. episcopus and T. sayaca, hence, “Systematics of Thraupis reveals hybridization of Thraupis episcopus (Blue-gray Tanager) at multiple scales” is not appropriated. It should be hybridization in the genus Thraupis, or between T. episcopus and T. sayaca.

- I suggest to the authors to explore more the impact of the hybridization between the Thraupis species. In the introduction they que cite other examples of hybridization among Passeriformes.

- Can you add in the discussion the divergence time between T. episcopus and T. sayaca?

Reviewer #3: This manuscript investigated the species boundaries of neotropical tanagers with plumage, mitochondrial and nuclear genetic analyses of hundreds of individuals across south America. This is a valuable study and I find the results very interesting with nice figures (though the resolution should be higher). However, the clarity of the writing can be largely improved and streamlined around the main taxonomic adjustment that T. episcopus and T. sayaca should be one species with at most five subspecies. Figures should really guide readers to inspect the plumage, mtDNA, and nuclear genetic evidence to reach this conclusion.

Major concerns:

First, he evidence that T. episcopus and T. sayaca should be a single species should be clearly presented with plumage, mtDNA and nuclear genetic evidence. The mtDNA evidence is almost clear, but authors should highlight the admixture between the species by labeling individuals sampled from T. episcopus and T. sayaca. The nuclear genetic evidence is unclear to me, please highlight this evidence on the phylogenetic tree.

Second, I am not convinced that there are at most five subspecies within this species complex. It depends on what K is the best fit to the data, and there are also possibly unsampled subdivisions.

Here are some minor concerns.

Title: vague, what scales?

Abstract:

(1) most people aren’t familiar with this species complex. Please highlight the part of your results that are inconsistent with the existing taxonomic understanding.

(2) Where is the evidence of population structure within T. episcopus? Did you mean episcopus-sayaca complex? These were already described as different taxonomic units?

Introduction, last sentence, “interspecific gene flow” should be hybridization.

Result

Please report best supported K.

Discussion

- “T. glaucocolpa is not related…” Here the “related” is vague. This shoud be “T. glaucocolpa is not sister species to”?

-

Fig.1 B-E, please label sayaca in the haplotype network. Since the purple cluster contains the green individuals, shouldn’t this be evidence against sayaca and episcopus being distinct species?

very hard to see cyan points on the map, consider either enlarge the dots, or change a color.

Fig.2 font problem, can’t see the text clearly. B, please use consistent color scheme as Fig. 1

Fig. 4, It’s a good idea to plot plumage color the map. The pink crosses are similar to the circles and squares in size, readers could be confused about the colors. Please increase the resolution.

Please try to combine Fig. 4 and Fig. 5, or should type specimens on Fig. 4.

6. PLOS authors have the option to publish the peer review history of their article (what does this mean?). If published, this will include your full peer review and any attached files.

Reviewer #1: **Yes: **Alparslan YILDIRIM

Reviewer #2: No

Reviewer #3: No

---

## [Author Response · Author response to Decision Letter 0]

3 Jun 2022

Louisiana State University

119 Foster Hall, Baton Rouge, LA 70803

April 2, 2021

Dear editor and reviewers,

Thank you for your time and comments reviewing this manuscript. We try to improve the manuscript as much as we could following your suggestions. Please follow the individual answers to each comment:

1. Please ensure that your manuscript meets PLOS ONE's style requirements, including those for file naming. The PLOS ONE style templates can be found at [links].

Thank you for drawing our attention to this point. Indeed, there were several things out of format. The format is now per PLoS One standards.

“DC

2015/22981-9 and 2017/03900-3

FAPESP – Fundação de Amparo à Pesquisa do Estado de São Paulo

https://fapesp.br/en

No

DC

No grant numbers

The Frank M. Chapman Memorial Fund of the American Museum of Natural History

https://www.amnh.org/research/vertebrate-zoology/ornithology/grants

No”

No funder had any role in the study design, data collection and analysis, decision to publish, or preparation of the manuscript. I add this statement in a new Financial Disclosure Statement. Additionally, we add some information form in the Financial Disclosure Statement.

3. We note that you have stated that you will provide repository information for your data at acceptance. Should your manuscript be accepted for publication, we will hold it until you provide the relevant accession numbers or DOIs necessary to access your data. If you wish to make changes to your Data Availability statement, please describe these changes in your cover letter and we will update your Data Availability statement to reflect the information you provide

Yes, I understand that you will hold my manuscript until I submit the sequences. Once the manuscript is accepted, I will upload the sequences to GenBank so accession number can be included in the manuscript.

4. We note that Figure 1, 2 and 4 in your submission contain map images which may be copyrighted. All PLOS content is published under the Creative Commons Attribution License (CC BY 4.0), which means that the manuscript, images, and Supporting Information files will be freely available online, and any third party is permitted to access, download, copy, distribute, and use these materials in any way, even commercially, with proper attribution. For these reasons, we cannot publish previously copyrighted maps or satellite images created using proprietary data, such as Google software (Google Maps, Street View, and Earth). For more information, see our copyright guidelines: http://journals.plos.org/plosone/s/licenses-and-copyright.

All images were produced by the authors, I included the following information on the figures with maps:

All maps in the paper were made using the free software Qgis v.3.10.7 and freedata from DIVA-GIS (https://www.diva-gis.org/).

Reviewer #1: The MS presents unique and novel daqta on the phylogenomics and the evaluation of the species in Thraupis genus. It also provides new insights in the distribution of the species and their boundaries with morphometric evaluations. Study was well design and the results were presented in detail. So I conclude that this MS could highly contribute to the current knowledge in the era of Thraupis.

Thank you.

Reviewer #2: In this study the authors investigated the systematics of Thraupis and the hybridization of Thraupis episcopus. The manuscript is very interesting and well-written. The study falls into the scope of Plos one, thus it could be published. Before this can be done, a minor revision should be made as specified below.

- I suggest changing the title. The authors found evidence of hybridization between T. episcopus and T. sayaca, hence, “Systematics of Thraupis reveals hybridization of Thraupis episcopus (Blue-gray Tanager) at multiple scales” is not appropriated. It should be hybridization in the genus Thraupis, or between T. episcopus and T. sayaca.

We accept the suggestion. We changed the title to “Systematics of Thraupis (Aves, Passeriformes) reveals an extensive hybrid zone between T. episcopus (Blue-gray Tanager) and T. sayaca (Sayaca Tanager)”

- I suggest to the authors to explore more the impact of the hybridization between the Thraupis species. In the introduction they que cite other examples of hybridization among Passeriformes.

Purposedly, we avoided talking deeply about hybridization because we are already working on a second paper where address this issue in further detail using genomic data (RADseq). Nonetheless, we decided to include a paragraph almost at the end of the introduction to mention some more information about hybridization in line 125.

Hybridization and introgression are important factors that affect evolution, diversification, and speciation processes. They can lead to different outcomes such as boostering speciation by reinforcement [22], merging taxa, generating new reticulate lineages [23,24] or even transferring advantageous alleles from one lineage to another [25]. However, the underlying mechanisms leading to either outcome remain partially unclear. Hybridization often occurs in nature between closely-related species, and it has been widely reported in various bird families within songbirds (Passeriformes) [26–31]. Currently, one of the reasons that can lead to hybridization is human-induced habitat transformation [32], which in the episcopus-sayaca complex might be relevant. Both species are largely parapatric and meet along the ecotone between Amazonia and the drier habitats in the Cerrado and Caatinga. Because Amazon limits are moving backward due to deforestation, the limits and interactions between species may have been changing.

Furthermore, we developed part of the discussion around the putative role of hybridization in driving the observed patterns.

- Can you add in the discussion the divergence time between T. episcopus and T. sayaca?

We added the information in the sentence that starts in line 542:

We found that T. episcopus and T. sayaca diverged ca.1.5 million years ago and currently hybridize over a broad region of central South America, along the ecotone between Amazonia and the South American Dry Diagonal, where they are parapatric (S12 Fig).

Reviewer #3: This manuscript investigated the species boundaries of neotropical tanagers with plumage, mitochondrial and nuclear genetic analyses of hundreds of individuals across south America. This is a valuable study and I find the results very interesting with nice figures (though the resolution should be higher). However, the clarity of the writing can be largely improved and streamlined around the main taxonomic adjustment that T. episcopus and T. sayaca should be one species with at most five subspecies. Figures should really guide readers to inspect the plumage, mtDNA, and nuclear genetic evidence to reach this conclusion.

Major concerns:

First, he evidence that T. episcopus and T. sayaca should be a single species should be clearly presented with plumage, mtDNA and nuclear genetic evidence. 

We do not propose that T. episcopus and T. sayaca should be a single species. On the contrary, on line 559 we explicitly state that “we suggest maintaining the species rank for T. episcopus and T. sayaca.” It is due to their morphological and ecological differences and the presence of a solid genetic structure. Eventhough there is a hybrid zone between these species in the Amazonia-Cerrado ecotone, it seems to be narrow and species integrities elsewhere are mainatined.

The mtDNA evidence is almost clear, but authors should highlight the admixture between the species by labeling individuals sampled from T. episcopus and T. sayaca. The nuclear genetic evidence is unclear to me, please highlight this evidence on the phylogenetic tree.

Trees in the supplemental material already have all individuals’ labels including museum where the sample is from, catalog number and species (same birds in Table 1). Moreover, individuals with mixed ancestry are already highlighted in magenta. The only tree without highlighted individuals was the BF5 tree (S2 Fig) because its low support across almost all the branches we do not consider it as evidence for the introgression. However, we remade FIG S2 highlighting hybrid individuals.

Second, I am not convinced that there are at most five subspecies within this species complex. It depends on what K is the best fit to the data, and there are also possibly unsampled subdivisions.

For the species complex (T. episcopus and T. sayaca), we recognize seven subspecies not five: T. episcopus episcopus, T. episcopus berlepschi, T. episcopus quaesita, T. episcopus cana, T. episcopus caesitia, T. sayaca sayaca and T. sayaca obscura. Thus, it is not clear to us what the comment of the reviewer refers to. However, we suppose that he refers to five subspecies within T. episcopus and the STRUCTURE plot in Fig 2B In that case, the information was not understood correctly. 

First, subspecies are discrete morphological geographic units that are not required to be reproductively isolated, otherwise they would be classified as species. Genetic structure and monophyly are not requirements for subspecific treatment, especially when analyzing few molecular markers that are not related with the phenotypically differences [1]. Therefore, we did not base our subspecific classification on molecular data only but rather on diagnosable phenotypic differences among named taxa within obtained genetic clusters. 

Second, we believe that results of the STRUCTURE plot (Fig 2B) is misinterpreted. It actually includes all individuals of both T. episcopus and T. sayaca and genetic clusters do not represent subspecies but rather groups of individuals with high levels of shared ancestry. These individuals represent different subspecies i.e., phenotypically diagnosable populations not reproductively isolated among them.

Finally, we are aware that we may have unsampled populations (it is not clear what means by subdivisions) that may alter the levels of resolution of our analyses. However, we would like to highlight that this is the most comprehensive phylogeographic study of the T. episcopus – T. sayaca to date and our sampling is an accurate representation of the distribution of these taxa. Future genomic analyses will provide further insights into the genetic structure of these populations.

Here are some minor concerns.

Title: vague, what scales?

The title has been modified following suggestions by reviewers 2 and 3.

Abstract:

(1) most people aren’t familiar with this species complex. Please highlight the part of your results that are inconsistent with the existing taxonomic understanding.

Thank you for this suggestion. We added the following sentences to the abstract: However, we found conflicts with previous phylogenies. We recovered Thraupis glaucocolpa to be sister to all other species in the genus, and T. cyanoptera to the remaining five species.

(2) Where is the evidence of population structure within T. episcopus? Did you mean episcopus-sayaca complex? These were already described as different taxonomic units?

The structure evidence can be seen in the phylogenetic and haplotype network analyses as well as on maps. Specifically, among the episcopus, cana and quaesita groups (Fig 1B and Fig 2A). These genetic groups are not necessarily taxonomic units formed by only one named taxon but rather represent clusters of several subspecies. We refer to each group using the senior synonym within each group.

To clarifu this point in abstract, we included the following sentence: “Our phylogenetic trees and population structure analyses uncovered phylogeographic structure within Thraupis episcopus that is congruent with geographic patterns of phenotypic variation and distributions of some named taxa.”

Introduction

last sentence, “interspecific gene flow” should be hybridization.

We replaced “interspecific gene flow” for “Introgressive hybridization” in line 123.

Results

Please report best supported K.

We reported the best supported K using the Evanno’s method. However, we made it more explicit in the sentence in line 391: “However, the Evanno et al.'s method suggests that the maximum number of individual populations is three (best supported K=3) [60]”

Discussion

- “T. glaucocolpa is not related…” Here the “related” is vague. This shoud be “T. glaucocolpa is not sister species to”?

We replaced “We found that T. glaucocolpa is not sister or closely-related to T. sayaca” in line 500

-Fig.1 B-E, please label sayaca in the haplotype network. Since the purple cluster contains the green individuals, shouldn’t this be evidence against sayaca and episcopus being distinct species?

We think Fig 1 B-E is clear. The color of the haplotypes is related to the name of the species in Fig 1 A. Purple haplotypes correspond to T. sayaca.

The green dots within the T. sayaca cluster (purple) and the purple dot within T. episcopus correspond to the individuals that are hybrids (exactly same individuals in Table 1). Some of these individuals had intermediate phenotypic characteristics. However, they were classified as T. episcopus because its appearance was closer to it. When mapped, these few birds were located in the contact zone between T. episcopus and T. sayaca in the Amazonia-Cerrado ecotone. A few hybrid individuals in a contact zone are not evidence against sayaca and episcopus being distinct species

very hard to see cyan points on the map, consider either enlarge the dots, or change a color.

Thank you for the comment. We suspect that the reviewer did not have access to the original file. The figure is high resolution and one-page size. All dots are easy to find, even the white ones.

Fig.2 font problem, can’t see the text clearly. B, please use consistent color scheme as Fig. 1

Thank you for the comment. We suspect the reviewer did not have access to the original file. The figure is high resolution and one-page size. We cannot use the same color schemes as in Fig 1. Because we are talking about different taxa. Fig 1 was current species within the genus Thraupis, while Fig 2 are the genetically structured population within Thraupis episcopus.

Fig. 4, It’s a good idea to plot plumage color the map. The pink crosses are similar to the circles and squares in size, readers could be confused about the colors. Please increase the resolution.

Thank you. We suspect the reviewer did not have access to the original file. The figure is high resolution and one-page size.

Please try to combine Fig. 4 and Fig. 5, or should type specimens on Fig. 4.

We tried to combine Fig 4 and 5 in different ways before the first submission. However, the figure looked overcharged and confusing. After trying different alternatives, we decided to include proceed with two separate figures. The geographic distribution of the plumages is better seen on Fig 1 and 2, S13 Fig and pictures in S1 folder.

Sincerely,

Diego Cueva, M.Sc.

Ph.D. Student – Louisiana State University

dacuevac@alumni.usp.br

---

## [Decision Letter · Decision Letter 1]

20 Jun 2022

Systematics of Thraupis (Aves, Passeriformes)reveals an extensive hybrid zone between T. episcopus (Blue-gray Tanager) and T. sayaca (Sayaca Tanager)

PONE-D-21-37811R1

Dear Dr. Cueva,

We’re pleased to inform you that your manuscript has been judged scientifically suitable for publication and will be formally accepted for publication once it meets all outstanding technical requirements.

Kind regards,

Tzen-Yuh Chiang

Academic Editor

PLOS ONE

Additional Editor Comments (optional):

Reviewers' comments:

Reviewer's Responses to Questions

**Comments to the Author**

1. If the authors have adequately addressed your comments raised in a previous round of review and you feel that this manuscript is now acceptable for publication, you may indicate that here to bypass the “Comments to the Author” section, enter your conflict of interest statement in the “Confidential to Editor” section, and submit your "Accept" recommendation.

Reviewer #1: All comments have been addressed

Reviewer #2: All comments have been addressed

2. Is the manuscript technically sound, and do the data support the conclusions?

Reviewer #1: Yes

Reviewer #2: Yes

3. Has the statistical analysis been performed appropriately and rigorously? 

Reviewer #1: Yes

Reviewer #2: Yes

4. Have the authors made all data underlying the findings in their manuscript fully available?

Reviewer #1: Yes

Reviewer #2: Yes

5. Is the manuscript presented in an intelligible fashion and written in standard English?

Reviewer #1: Yes

Reviewer #2: Yes

6. Review Comments to the Author

Reviewer #1: The quality of the MS is now increased by addressing each comment indicated by reviewers. I think the MS could be published in PLOS One.

Reviewer #2: Dear Authors, I am happy with the new version of your manuscript. It can be accepted for publication now.

Best wishes

7. PLOS authors have the option to publish the peer review history of their article (what does this mean?). If published, this will include your full peer review and any attached files.

Reviewer #1: No

Reviewer #2: No

---

## [Editor Report · Acceptance letter]

26 Sep 2022

PONE-D-21-37811R1 

Systematics of *Thraupis* (Aves, Passeriformes) reveals an extensive hybrid zone between *T. episcopus* (Blue-gray Tanager) and *T. sayaca* (Sayaca Tanager) 

Dear Dr. Cueva:

I'm pleased to inform you that your manuscript has been deemed suitable for publication in PLOS ONE. Congratulations! Your manuscript is now with our production department. 

Kind regards, 

on behalf of

Dr. Tzen-Yuh Chiang 

Academic Editor

PLOS ONE